# Enhancing Rearing of European Seabass (*Dicentrarchus labrax*) in Aquaponic Systems: Investigating the Effects of Enriched Black Soldier Fly (*Hermetia illucens*) Prepupae Meal on Fish Welfare and Quality Traits

**DOI:** 10.3390/ani13121921

**Published:** 2023-06-08

**Authors:** Matteo Zarantoniello, Adriana Alves de Oliveira, Tolga Sahin, Lorenzo Freddi, Matteo Torregiani, Isabella Tucciarone, Giulia Chemello, Gloriana Cardinaletti, Elia Gatto, Giuliana Parisi, Cristiano Bertolucci, Paola Riolo, Ancuta Nartea, Giorgia Gioacchini, Ike Olivotto

**Affiliations:** 1Department of Life and Environmental Sciences, Marche Polytechnic University, 60131 Ancona, Italy; m.zarantoniello@univpm.it (M.Z.); s1101273@studenti.univpm.it (M.T.); g.chemello@univpm.it (G.C.); giorgia.gioacchini@univpm.it (G.G.); 2CIMAR/CIIMAR Interdisciplinary Centre of Marine and Environmental Research, University of Porto, 4450-208 Matoshinos, Portugal; aoliveira@ciimar.up.pt; 3Department of Aquaculture, Marine Sciences and Technology Faculty, Çanakkale Onsekiz Mart University, 17000 Çanakkale, Turkey; tolgasahin@comu.edu.tr; 4Mj Energy srl Società Agricola, Contrada SS. Crocifisso, 22, 62010 Treia, Italy; lfreddi93@gmail.com; 5Department of Agriculture, Food, Environment and Forestry, University of Florence, 50144 Firenze, Italy; isabella.tucciarone@unifi.it (I.T.); giuliana.parisi@unifi.it (G.P.); 6Department of Agricultural, Food, Environmental and Animal Sciences, University of Udine, 33100 Udine, Italy; gloriana.cardinaletti@uniud.it; 7Department of Chemical, Pharmaceutical and Agricultural Sciences, University of Ferrara, 44100 Ferrara, Italy; gttlei@unife.it; 8Department of Life Sciences and Biotechnology, University of Ferrara, 44100 Ferrara, Italy; bru@unife.it; 9Department of Agricultural, Food and Environmental Sciences, Polytechnic University of Marche, 60131 Ancona, Italy; p.riolo@staff.univpm.it (P.R.); a.nartea@univpm.it (A.N.)

**Keywords:** histology, gut health, aquaponics, immune response, fish behavior

## Abstract

**Simple Summary:**

Modern aquaculture is moving towards sustainable aquafeed formulations and farming techniques that allow the application of the circular economy concept and the reduction of water consumption. In this regard, the present study is focused on the use of black soldier fly full-fat prepupae meal (HPM) enriched with spirulina as a more sustainable aquafeed ingredient in diets intended for European seabass reared in aquaponic systems. The results obtained from the present study allowed us to demonstrate the suitability of aquaponic systems for the rearing of euryhaline finfish species and the beneficial properties of spirulina-enriched HPM on fish growth, welfare, and quality features.

**Abstract:**

Within the modern aquaculture goals, the present study aimed to couple sustainable aquafeed formulation and culturing systems. Two experimental diets characterized by 3 and 20% of fish meal replacement with full-fat spirulina-enriched black soldier fly (*Hermetia illucens*) prepupae meal (HPM3 and HPM20, respectively) were tested on European seabass (*Dicentrarchus labrax*) juveniles during a 90-day feeding trial performed in aquaponic systems. The experimental diets ensured 100% survival and proper zootechnical performance. No behavioral alterations were evidenced in fish. Histological and molecular analyses did not reveal structural alterations and signs of inflammation at the intestinal level, highlighting the beneficial role on gut health of bioactive molecules typical of HPM or derived from the enriching procedure of insects’ growth substrate with spirulina. Considering the quality traits, the tested experimental diets did not negatively alter the fillet’s fatty acid profile and did not compromise the fillet’s physical features. In addition, the results highlighted a possible role of spirulina-enriched HPM in preserving the fillet from lipid oxidation. Taken together, these results corroborate the use of sustainable ingredients (spirulina-enriched HPM) in aquaponic systems for euryhaline fish rearing.

## 1. Introduction

The human population is expected to reach 8.5 billion by 2030 and 9.7 billion by 2050 [1], raising serious concerns about future food supply. In this regard, the aquaculture industry plays a key role, and it is projected that by 2030, about 62% of seafood will be produced through aquaculture [2]. Nevertheless, aquaculture must meet a variety of sustainability criteria, such as reducing organic pollution and water consumption, while maintaining high production standards for product quality and animal welfare, meeting the European Union requirements [3]. For these reasons, research must focus on both aquafeed formulation and culturing systems, potentially contributing to the Sustainable Development Goals of the United Nations (SDG) [4] and following the FAO principles towards a sustainable future of aquaculture with a focus on the concept of the circular economy [5].

Regarding aquafeed production, several ingredients have been investigated as promising alternatives to replace conventional and less sustainable marine-derived ingredients, such as fish meal (FM) and oil [6], including plant-based ingredients [7,8], processed animal proteins [9,10], protein sources derived from aquatic invertebrates [11,12], and microbial biomasses [13,14,15].

Among the alternatives, insects have recently gained increasing attention due to their many beneficial properties [16]. Insects have a low environmental footprint since they can be reared on waste organic by-products with low water input, emitting low levels of greenhouse gases and ammonia [17]. Furthermore, insect rearing can contribute to waste biodegradation by converting low-value organic waste into high-value ingredients, which allows the application of the circular economy concept to the aquaculture sector [18,19]. In addition, insects exhibit high feed conversion ratio, fast growth, and reproduction rates [20], which enables commercial-scale production [21]. Moreover, based on the regulation adopted by the European Union, the use of processed animal protein from insects in aquafeed has been authorized, and a number of species, including the black soldier fly (*Hermetia illucens*), have been approved when reared using feed-grade substrates [22].

From a nutritional standpoint, the high protein content (around 60% or up to 65.5% in defatted larvae), well-balanced essential amino acid profile similar to FM, and the vitamin and mineral content makes the black soldier fly prepupae meal (HPM) an apparent suitable option for inclusion as ingredient in aquafeed formulations [23,24]. Furthermore, HPM is rich in bioactive molecules such as chitin, lauric acid, and antimicrobial peptides, which have been reported to have immunostimulatory and anti-inflammatory effects on fish gut [16]. Nevertheless, the HPM fatty acid profile does not meet the fish nutritional requirements, lacking in long-chain polyunsaturated fatty acids (PUFA), such as EPA and DHA, which raises serious concerns about the use of full-fat HPM in feed formulations, especially for marine fish species [25,26]. For this reason, several studies have been conducted with defatted HPM in both marine and freshwater fish species [27,28,29,30]. However, the desirable goal of the aquaculture industry is to utilize both proteins and lipids from insect biomass.

In this regard, black soldier fly larvae can modulate their fatty acid profile according to the composition of their feeding substrate [31,32]. In this context, it has been demonstrated that the microbial biomass of marine protists and cyanobacteria, characterized by the presence of long-chain PUFA and bioactive molecules, can be a valid option to improve HPM’s nutritional value [33,34,35].

Spirulina (*Arthrospira platensis*), the most commercially produced species of cyanobacteria among phytoplankton, contains PUFA and numerous bioactive compounds comprising phycobiliproteins, carotenoids, phycocyanin, polysaccharides, and a number of vitamins, which have the ability to positively affect fish health and quality [36,37,38]. In this regard, recent studies demonstrated that the inclusion of spirulina in the insect growth substrate was able to increase both long-chain PUFA and antioxidant molecules content in HPM and, consequently, in the diets intended for rainbow trout (*Oncorhynchus mykiss*) and giant freshwater prawn (*Macrobrachium rosenbergii*) culture, leading to positive physiological responses in both these species [35,39].

In an ever-growing aquaculture, aquafeed formulations, together with water consumption, represent the key factors challenging the long-term sustainability of this sector. For this reason, this highly productive sector has to move towards more sustainable practices, placing emphasis on the circular economy concept [4,5]. Accordingly, the present study coupled the use of a sustainable aquafeed ingredient with a sustainable culturing system. Specifically, two different dietary replacement levels (3 and 20%) of FM with full-fat spirulina-enriched HPM (according to previous studies [35,39]) were tested during a 90-day feeding trial performed on European seabass juveniles (*Dicentrarchus labrax*), an economically relevant marine species in the European Union aquaculture [40]. The trial was performed in aquaponic systems, which: (i) can reduce water consumption by over 90% compared to traditional agriculture and aquaculture, growing in importance especially in areas with limited water supply; and (ii) promote the reuse of organic by-products in line with a waste-free aquaculture development, reducing the environmental impact associated with effluent discharge [41]. At the end of the trial, a multidisciplinary methodology, encompassing zootechnical parameters, marketable quality traits, and physiological and behavioral responses, was used to assess fish quality and welfare.

## 2. Materials and Methods

### 2.1. Ethics

All protocols and procedures for fish manipulation were performed according to the guidelines established by the Council of the European Communities Directive (86/609/EEC and 2010/63/EU) for animal welfare and approved by the Animal Ethics Committee of Marche Polytechnic University (n. 01/2022-UT of 17 January 2022). Optimal rearing conditions were ensured, and animal suffering was minimized using the MS222 anesthetic (Merck KGaA, Darmstadt, Germany).

### 2.2. Insects’ Rearing and Production of Fish Diets

Insects’ rearing was conducted following the procedures reported in Ratti et al. [35]. Briefly, the rearing substrate was mainly composed of coffee industry by-product (coffee silverskin) in which 15% (*w*/*w*) of the dried biomass from Arthrospira platensis was included, reaching a final moisture of ~70% by adding distilled water. The black soldier fly larvae (provided by Smart Bugs s.s., Ponzano Veneto, Treviso, Italy) were reared on the feeding substrate placed in a climatic chamber set at 27 ± 1 °C and 65 ± 5% of relative humidity. Larvae were reared at a 0.3/cm^2^ density with a feeding rate of 100 mg/day per larva [42,43], ensured by adding weekly new feeding substrate. Insects’ rearing was performed in continuous darkness for 21 days, until the prepupal stage was reached (identified by the transition in tegument color from white to black). Prepupae were then washed, freeze-dried and preserved at −80 °C until being ground to obtain fine powder (Retsch Centrifugal Grinding Mill ZM 1000, Retsch GmbH, Haan, Germany) for fish diet preparation.

Three experimental diets were formulated to cover the nutritional requirements of European seabass [44]. Specifically, a control diet (HPM0) based on conventional marine (FM and fish oil) and vegetable ingredients, and two diets obtained from the control diet by replacing FM at 3% (as a feed supplement inclusion) or 20% (this dietary FM replacement level in aquafeeds at the global level has a significant ecological impact) with spirulina-enriched HPM (named HPM3 and HPM20, respectively) were produced. To prepare the test diets, all dried ground ingredients were thoroughly blended (GastroNorm 30C1PN, ItaliaGroup Corporate Srl, Ponte nelle Alpi, Italy) for 20 min and water was then added to the mixture to attain an appropriate consistency for pelleting. Pellets were obtained by using a 3 mm die meat grinder, dried at 37 °C for 48 h in a ventilated heater, and then ground, sieved, and stored in vacuum bags and kept at −20 °C until use.

The proximate composition of the experimental diets was analyzed by following AOAC (2016) procedures [45], while the proximate composition of HPM has been reported by Ratti et al. [35]. The measure of moisture content was obtained through weight loss after sample drying in an oven (105 °C), until a constant weight was reached. Ash content was evaluated via incineration in a muffle furnace by combustion at 550 °C for 4 h (HD 230, AMSE s.r.l.; Torino, Italy). Crude protein was determined as total nitrogen (N) through the Kjeldahl method. The nitrogen-to-protein conversion factor of 6.25 was used for protein content estimations of the experimental diets. Crude fat was determined according to the Soxhlet ether method. Finally, the gross energy was measured by a calorimetric bomb (Adiabatic Calorimetric Bomb Parr 1261; PARR Instrument, IL, USA).

For the fatty acid profile, a base-catalyzed transesterification of 4 mg total lipid in methyl esters (FAME) [46] was used; FAME was then injected into a Varian GC gas chromatograph (Varian Inc, Palo Alto, CA, USA) equipped with a flame ionization detector (FID) and a Supelco OmegawaxTM 320 capillary column (30 m, 0.32 mm i.d, 0.25 μm film, and polyethylene glycol-bonded phase; Supelco, Bellefonte, PA, USA). The injector and detector temperatures were set at 220 °C and 300 °C, respectively. The carrier gas was helium (constant flow of 1.5 mL/min). Chromatograms were recorded with the Galaxie Chromatography Data System 1.9.302.952 (Varian Inc., Palo Alto, CA, USA). Fatty acids were identified by comparing the retention times of FAMEs with those of the standard Supelco 37 component FAME mix (Supelco, Bellefonte, PA, USA) and quantified through calibration curves, using tricosanoic acid (C23:0) (Supelco, Bellefonte, PA, USA) as an internal standard. Data were expressed as a percentage of the total FAMEs.

For the α-tocopherol and carotenoid determination, the protocols described by Ratti et al. [35] and Nartea et al. [47], respectively, were adopted. Briefly, for α-tocopherol determination, freeze-dried samples (in triplicate) were added to 5 mL of hexane, vortexed for 5 min, and centrifuged at 3500 rpm for 2 min. For carotenoids, freeze-dried samples (in triplicate) were added to acetone (5 mL, 4 °C), kept at 4 °C for 15 min, vortexed for 5 min, and then centrifuged (1370 rpm, 10 min, 4 °C). For both α-tocopherol and carotenoids, the organic fraction was collected, filtered (0.45 μm, Sartorius Regenerated Cellulose Membrane), dried, resuspended in 0.5 mL hexane or 0.5 mL acetone, respectively, and then injected into a Waters Ultra Pressure Liquid Chromatographic Acquity system (UPLC Acquity H-Class, Waters Corporation, Milford, CT, USA). The UPCL system was equipped with a Fluorimetric Detector (FLD) and an Ascentis Express HILIC column (15 cm × 2.1 mm, 2.7 μm) for α-tocopherol or with a Photodiode Array Detector (PDA) for carotenoids. Both α-tocopherol and carotenoids were identified by comparison of retention time and absorbance spectrum with pure standards. Good correlation coefficients (R^2^) of 0.999 were obtained in all cases.

Table 1 shows the ingredients, the proximate composition (% as fed), and the α-tocopherol and carotenoid contents of the experimental diets used in the present study.

### 2.3. Experimental Design

The feeding trial was conducted at Mj Energy srl Società Agricola (Treia, Macerata, Italy). Two hundred and seventy European seabass juveniles, provided by Panittica Pugliese (Torre Canne di Fasano, Brindisi, Italy), were acclimated to the experimental conditions for two weeks in a single 1000 L tank equipped with mechanical, biological, and UV filtration (Panaque s.r.l., Viterbo, Italy). Water parameters were as follows: temperature, 17.0 ± 0.5 °C; salinity 30 ± 0.5‰; ammonia and nitrite < 0.05 mg/L; nitrate < 10 mg/L. During the acclimation period, salinity was gradually decreased from 30 to 5‰ (2‰ daily by adding freshwater) to match the salinity of aquaponic systems. At the end of the acclimation period, fish (with an initial body weight of 19.3 ± 0.1 g) were randomly divided into nine Media-Based Aquaponic Systems (600 L fish tank and 120 L hydroponic unit), with three systems per experimental group (30 specimens per tank). Fish were subjected to a natural photoperiod (13L/11D) and water in the aquaponic systems was maintained at a constant temperature (17.0 ± 0.2 °C) by TK500 chillers (Teco, Ravenna, Italy) and at 5 ± 0.5‰ of salinity. A 1900 L/h pump (Eheim GmbH & Co, Deizisau, Germany) regulated the water flow, completing 3 water renewals each hour. From each fish tank, water was pumped to the corresponding hydroponic unit, going back through a siphon in which synthetic foam was added to provide extra mechanical filtration. In each hydroponic unit, 100 saffron (Crocus sativus) seedlings were planted, at a density of 65 plants/m^2^, one day before the introduction of fish in the aquaponic systems. Each hydroponic unit was equipped with expanded clay to ensure physical support for plant growth and both mechanical and biological filtration for fish tanks.

Water samples were collected once a week to monitor ammonia, nitrite, nitrate, and phosphate concentrations (Hanna reagents and HI83399 spectrophotometer; Hanna instruments, Villafranca Padovana, Italy). The duration of the feeding trial was 90 days, during which fish almost triplicated their weight. Fish were hand-fed the experimental diets at 2% body weight, divided into one ration in the morning and one in the afternoon. The daily feed was adjusted every two weeks by weighing a representative number of specimens for each tank. Fish from the HPM0 group were fed the control diet, while fish from the HPM3 and HPM20 groups were fed diets characterized by 3 or 20% of FM replacement with HPM, respectively. The feed particle size was 3 mm. At the end of the trial, all the fish from each tank were euthanized with a lethal dose of anesthetic (MS222, 0.3 g/L), then counted and individually weighted. The whole fish and their fillets, as well as samples from the liver and distal intestine, were collected and properly stored for further analyses.

### 2.4. Survival Rate and Zootechnical Performance

For each tank, survival rate (SR), relative growth rate (RGR), specific growth rate (SGR), and feed conversion ratio (FCR) were calculated as follows:SR (%) = (final number of fish/initial number of fish) × 100(1)
RGR (%) = [(FBW − IBW)/IBW] × 100(2)
SGR (% day^−1^) = [(ln FBW − ln IBW)/days of trial] × 100(3)
FCR = feed intake per tank/weight gain per tank(4)
where FBW and IBW are final and initial body weights, respectively.

### 2.5. Behavioral Test

The behavioral test was performed one week before the end of the trial. Fish exploration tendency in a novel environment was individually assayed using an open-field test [48,49,50]. A subsample of 4 fish per tank (12 fish per each experimental condition) was individually collected with a net and gently transported to the experimental arena. The arena consisted of a plastic rectangular tank (75 × 75 × 30 cm). The long walls of the arena were white, while the bottom was light gray. The arena was positioned into a larger tank (120 × 120 × 50 cm) connected to a recirculating system with the same water features of the aquaponic systems.

The behavior of subjects was recorded for 20 min with a camera (Sony Handycam HDR-CX405; resolution camera: 1920 × 1080; 25 frames per second) placed 150 cm above the arena. Behavioural performance from the video recordings was analyzed with EthoVision XT software (ver. 11.5, Noldus Information Technology, Wageningen, The Netherlands) which allowed us to track the fishes’ swimming behavior in any virtual zones of the arena. Different individual movement variables of interest were chosen to analyze the fishes’ explorative behavior [51,52,53]: (i) the distance covered by each fish in the tank (in cm); (ii) the time spent exploring the center zone (20 × 20 cm) of the arena; (iii) the absolute turn angle (in °) during fish movement, as the difference in angular changing in swimming direction between two successive samples, used as an indicator of anxious phenotype; and (iv) immobility state. This variable was calculated as the 20% or below difference in the degree of movement changing in an interval of 5 samples.

In addition, breathing frequency was measured as an indicator of stress level [54,55]. The video recordings were played back using a computer to manually count the number of gill movements and consequently calculated an index as follows:∆breathing frequency = gill movement during the last 2 min − first 2 min of the test(5)

At the end of the test, all the fish were transferred to their original aquaponic system.

### 2.6. Physical and Chemical Parameter Analyses

#### 2.6.1. Physical Analyses and Marketable Indexes

Eight samples from each experimental group were subjected to physical parameter analyses. Specifically, each sample consisted of 2 fish (16 fish per experimental group in total) which were then identified as A and B. The fish were taken out of the −80 °C freezer where they were stored and kept overnight at a temperature of 1 °C to allow for slow thawing. After measuring the total and muscle lengths (standard length), each fish was weighed and the color of the skin on the fish’s left side was determined. Each fish was then dissected and the weight and color of both fillets (right and left) were recorded. The color of skin and fillets was measured in three positions (cranial, medial, and caudal), with a CHROMA METER CR-400/410 (Konica Minolta, Chiyoda, Japan) recording the color parameters L* (lightness), a* (redness index), and b* (yellowness index). After that, the following parameters were also calculated:Condition factor, K (%) = [(body weight (g)/total length (cm)^3^) × 100(6)
Fillet yield, FY (%) = [(fillet weight with skin (g)/body weight (g)] × 100(7)

Finally, the fillets were skinned and homogenized for chemical analyses.

#### 2.6.2. Lipid Extraction and Fatty Acid Identification

For the chemical analyses, because of the fish size, the minced fillets of each specimen A were combined with the minced fillets of the corresponding specimen B in order to obtain a single sample. The total lipids of fillets were determined. Lipids were extracted according to Folch et al.’s [56] method, starting, for each sample, from 2 g of minced fillet. The total lipids were therefore quantified gravimetrically and expressed as g of lipids/100 g of fillet. The fatty acid profile was determined with the same procedure described above for the diets.

#### 2.6.3. Oxidative Status

To determine the oxidative state of the fillets, primary and secondary oxidative products were quantified as conjugated dienes (CD) and thiobarbituric acid reactive substances (TBARS), respectively. First, CDs were quantified by a spectrophotometric method by dissolving 0.1 μL of lipid extract in 3 mL of pure hexane. Then, using a Lambda spectrophotometer EZ 150 UV/VIS (Spectrometer, Perkin Elmer, Waltham, MA, USA), the absorbance of the sample was detected at 232 nm. For the calculation of the conjugated dienes, a molar extinction coefficient of 25,200 mL/mmol was used and the results were expressed as the mmol Hp/100 g of sample. Subsequently, 2 g of homogenized fillets were utilized to determine lipid oxidation products (TBARS). TBARS were determined according to the spectrophotometric method previously proposed by Vyncke [57], which quantifies the reactive substances of thiobarbituric acid (TBARS) by measuring the equivalent malondialdehyde (MDA eq.). The reading with the Lambda EZ 150 UV/VIS spectrophotometer (Perkin Elmer, Waltham, MA, USA) was made at 532 nm and the results were expressed as mgMDA-eq./100 g of fresh sample.

### 2.7. Histological Analyses

For each tank, liver and distal intestine samples obtained from 10 fish (30 fish per experimental group) were fixed in Bouin’s solution for 24 h at 4 °C and subsequently washed in 70% ethanol. Samples were dehydrated using 80, 95, and 100% ethanol solution, washed with xylene, and finally embedded in paraffin (Bio-Optica, Milano, Italy) [58]. Paraffin blocks were cut to obtain sections of 5 µm (using a RM2125 RTS microtome; Leica, Nussloch, Germany) that, after staining, were examined using a Zeiss Axio Imager.A2 microscope equipped with an Axiocam 105 combined color digital camera (Zeiss, Oberkochen, Germany).

The evaluation of distal Intestine morphology and the assignment of histopathological indexes were performed on three transversal sections per fish (30 fish per experimental group), taken at 200 μm intervals and stained with: (i) Mayer’s hematoxylin and eosin Y (Merck kGaA, Darmstadt, Germany) for mucosal fold height, submucosa width, mucosal fold fusion, basal inflammatory influx, and supranuclear vacuolization of enterocytes; and (ii) Alcian blue (Bio-Optica, Milano, Italy) for the relative abundance of Alcian blue positive (Ab+) goblet cells calculated on a 500 µm^2^ surface area. The score assignment criteria for the above-mentioned parameters are reported in Table 2. The image analysis was performed using the ZEN 2.3 software (Zeiss, Oberkochen, Germany).

Three liver sections per fish (30 fish per experimental group) were taken at 200 μm intervals and stained with: (i) Mayer’s hematoxylin and eosin Y (Merck kGaA, Darmstadt, Germany) to assess hepatocyte morphology and the degree of fat accumulation; and (ii) Periodic Acid Schiff (PAS; Bio-Optica, Milano, Italy) to highlight glycogen deposition in hepatocytes. Regarding the degree of fat accumulation, a quantitative analysis was performed on three sections per liver sample (30 fish per experimental group). Blood vessels and bile ducts were excluded from quantitative analysis since they are considered non-assessable areas. Results were expressed as the percentage of area occupied by fat on the total hepatic parenchyma analyzed in the respective section.

Image analysis was performed using the ZEN 2.3 software (Zeiss, Oberkochen, Germany), while the percentage of fat fraction was determined with ImageJ.

### 2.8. Spectroscopic Analyses

For each tank, liver samples obtained from 5 fish (15 fish per experimental group) were collected and stored at −80 °C for Fourier Transform Infrared Imaging (FTIRI) analysis. A Bruker Invenio interferometer coupled with a Hyperion 3000 Vis-IR microscope and equipped with a Focal Plane Array (FPA) detector operating at liquid nitrogen temperature (Bruker Optics, Ettlingen, Germany) was used for FTIRI measurements. Three sections of 10 µm, taken at 200 µm intervals using a cryotome (MC4000, Histo-Line Laboratories, Milano, Italy), were deposited on CaF_2_ optical windows (1 mm thick, 13 mm diameter) and allowed to air-dry for 30 min. Using a 15× condenser/objective, specific areas were detected on each section, on which infrared maps of 164 × 164 µm size (4096 pixel/spectra with a spatial resolution of 2.56 × 2.56 μm) were acquired in transmission mode in the mid-infrared range (4000–800 cm^−1^; 4 cm^−1^ of spectral resolution; 256 scans). The background spectrum was acquired on a clean portion of the optical window before each sample acquisition. The Atmospheric Compensation (for the correction of CO_2_ and water vapor atmospheric contributions) and the Vector Normalization (applied on the full frequency range to avoid thickness variations) routines (OPUS 7.5 software package; Bruker Optics, Ettlingen, Germany) were used to pre-process all the raw infrared maps.

To evaluate the biochemical composition of hepatic tissue, false-color images were obtained from the integration process of infrared maps in the 2995–2820 cm^−1^ (representative of CH_2_ and CH_3_ groups in lipid chains, LIP), 1755–1720 cm^−1^ (representative of fatty acid regions, FA), and 1082–960 cm^−1^ (representative of glycogen, GLY).

For each IR map, the integrated areas of the spectral ranges mentioned above were also calculated and used to obtain the following band area ratios: LIP/TBM, FA/LIP, and GLY/TBM; TBM represents the sum of the integrated areas calculated in the 3050–2800 cm^−1^ and 1800–900 cm^−1^ spectral ranges.

### 2.9. Molecular Analyses

Total RNA extraction from distal intestine samples obtained from 5 fish per tank (15 fish per experimental group) was performed using TRI Reagent (Merck kGaA, Darmstadt, Germany). The final RNA concentration was obtained with a Nanophotometer P-Class (Implen, München, Germany), while the RNA integrity was checked by 28S and 18S ribosomal RNA band staining (GelRed™) on 1% agarose gel. The cDNA synthesis was performed with the iScript™ cDNA synthesis kit (Bio-Rad, Hercules, CA, USA) using 1 μg of total RNA.

Real-time qPCR were performed in an iQ5 iCycler thermal cycler (Bio-Rad, Hercules, CA, USA) using a 96-well plate according to Chemello et al. [59]. For each sample, 1 μL of 1:10-diluted cDNA, 5 μL of fluorescent intercalating agent (2× concentrated iQ™ Sybr Green, Bio-Rad, Hercules, CA, USA), 0.3 μM of forward primer, and 0.3 μM of reverse primer were mixed. The thermal profile was set as follows: 3 min at 95 °C and then 45 cycles of 20 s at 95 °C, 20 s at the specific annealing temperature of each primer (reported in Table 3), and 20 s at 72 °C. For all the qPCRs performed, the melting curve analyses showed one single peak, no amplification products were observed in negative controls, and control templates did not highlight primer-dimer formation. Relative quantification of the expression of genes involved in immune response (Toll-like receptor 1, tlr1; myeloid differentiation primary response, myd88; nuclear factor kappa-light-chain-enhancer of activated B cells, nfkb; interleukin-1β, il1b; interleukin-10, il10; tumor necrosis factor alpha, tnfa) was performed. Beta-actin (β-actin) and 18S ribosomal protein (18s) were used as housekeeping genes (hk). The primer sequences were retrieved from NCBI (http://www.ncbi.nlm.nih.gov, accessed on 9 February 2023) and are summarized in Table 3. Amplification products were sequenced, and homology was verified. The iQ5 optical system software version 2.0 (Bio-Rad, Hercules, CA, USA), including GeneEx Macro iQ5 Conversion and GeneEx Macro iQ5 files, was used to analyze PCR data.

### 2.10. Statistical Analyses

The aquaponics systems were used as the experimental unit for data related to survival rate and zootechnical performance, while fish were considered the experimental unit for all the remaining analyses. All data were checked for normality (Shapiro–Wilk test) and homoscedasticity (Levene’s test). Data on ∆breathing frequency, fillet quality, histological analysis, spectroscopic analyses, and relative quantification of gene expression were analyzed using a one-way analysis of variance (ANOVA) followed by a Tukey’s multiple comparison test. Significance was set at *p* < 0.05, and results were expressed as mean ± standard deviation (SD). Data on explorative behavior (i.e., distance covered, immobility, time exploring the center of the arena, and turn angle) did not meet the assumption of normality and/or homoscedasticity. The non-parametric Kruskal–Wallis test was used for analyzing each set of behavioral data and the results were expressed as first quantile (Q1)—median—and third quantile (Q3). The statistical software package Prism-8 (GraphPad Software version 8.0.2, San Diego, CA, USA) was used for the data analysis.

## 3. Results

### 3.1. Water Quality Parameters

In all the experimental groups, ammonia and nitrite concentrations were lower than 0.05 mg/L at each measurement. No significant differences were observed in terms of nitrate (82.0 ± 1.8, 79.0 ± 1.0, and 80.1 ± 1.0 mg/L for HPM0, HPM3, and HPM20, respectively) and phosphate (1.4 ± 0.1, 1.4 ± 0.1, and 1.3 ± 0.1 mg/L for HPM0, HPM3, and HPM20, respectively) concentrations among the experimental groups.

### 3.2. Survival Rate and Zootechnical Performance

At the end of the feeding trial, the survival rate (SR) of the fish was 100% in all the experimental groups and all the feed provided was completely consumed by fish in all experimental groups. Zootechnical performance is reported in Table 4. Final body weight (FBW), relative growth rate (RGR), specific growth rate (SGR), and feed conversion ratio (FCR) did not show significant differences among the experimental groups.

### 3.3. Behavioral Test

The analysis of each behavioral measurement during the open-field test, i.e., distance covered, immobility, time exploring the center of the arena, mean turn angle, and ∆breathing frequency, did not show significant differences among the experimental groups (Table 5).

### 3.4. Physical and Chemical Analyses

#### 3.4.1. Experimental Diets

Table 6 shows the fatty acid profiles of the experimental diets. The increasing HPM dietary inclusion resulted in an increase in saturated fatty acid (SFA) and a parallel reduction in n3 PUFA. Differently, considering monounsaturated fatty acid (MUFA) and n6 PUFA, the experimental diets showed comparable values. As regards the fatty acid profile, the palmitic (C16:0) and the oleic (C18:1n9) acids were the most represented SFA and MUFA, respectively, in all the experimental diets. The increasing HPM dietary inclusion led to an increase in lauric acid (C12:0) and a parallel slight reduction in α-linolenic (C18:3n3), arachidonic (C20:4n6), eicosapentaenoic (C20:5n3; EPA), and docosahexaenoic (C22:6n3; DHA) acids.

#### 3.4.2. Physical Parameter Analyses

The physical characteristics of European seabass juveniles were not significantly affected by dietary treatments, except for the fillet yield, the intestine/spleen weight, and the lightness index (for both skin and fillet) (Table 7). First, the fillet yield of HPM0 and HPM3 groups were similar and significantly different from the HPM20 one (*p* < 0.05). In addition, the weight (and percentage) of the intestine and spleen together was also significantly different between the HPM0 and HPM20 groups (*p* < 0.05). The skin lightness (L*) of the HPM0 group was higher than that of the other experimental groups. On the contrary, the highest fillet lightness value was evidenced by the HPM20 group. Finally, the HPM3 group showed the lowest (*p* < 0.05) lightness values of both skin and fillet in comparison with the other experimental groups.

#### 3.4.3. Total Lipids and Fatty Acid Profile

Table 8 reports the total lipids and the fatty acid profiles of fish fillets. No significant differences were detected in total lipids and in both MUFA and n3 PUFA content among the experimental groups. The HPM20 group showed significantly (*p* < 0.05) lower SFA content compared to the other experimental group, while the HPM3 group was characterized by a significantly (*p* < 0.05) higher n6 PUFA content compared to the HPM0 and HPM20 groups. As regards the fatty acid profile of the fillets, the HPM20 group showed a significantly (*p* < 0.05) lower content of palmitic (C16) and stearic (C18:0) acid. Finally, the linoleic acid (C18:2n-6) content was significantly (*p* < 0.05) higher in the HPM3 group compared to both the HPM0 and HPM20 groups.

#### 3.4.4. Oxidative Status

The results of lipid oxidation (Table 9) revealed that conjugated dienes (CD) and TBARS content of fish fillets were not significantly affected by diets.

### 3.5. Histological Analyses

Figure 1 shows an example of distal intestine histomorphology of the samples obtained in the present study and highlights the measurement criteria for the mucosal fold height and submucosa width, as well as the presence of Ab + goblet cells.

Distal intestine samples from all the experimental groups did not show major signs of inflammation or alterations of histological architecture (Figure 2a–c). The morphometric evaluations and histological index scores are reported in Table 10. No significant differences were evident in terms of mucosal fold height and submucosa width among the experimental groups that were equally characterized by isolated events of mucosal fold fusion and by the absence of enterocytes’ supranuclear vacuolization. Finally, fish from the HPM20 group showed a higher abundance of Ab + goblet cells compared to those from the HPM0 and HPM3 groups (Figure 2d–f).

Considering the liver, all the experimental groups were characterized by a normal structure of hepatic parenchyma and by the absence of appreciable inflammation signs. However, the results evidenced a variable degree of lipid accumulation in all the experimental groups. The HPM0 and HPM3 groups were characterized by a modestly fat liver parenchyma with a diffuse presence of hepatocytes and cytoplasm filled with fat, interspersed with normal hepatocytes (Figure 3a,b). Conversely, liver samples from the HPM20 group showed a more severe degree of fat accumulation, with swollen hepatocytes and abundant intracytoplasmic lipid accumulation (Figure 3c). These results were confirmed by the statistical quantification of fat fraction percentage on liver sections: the HPM20 group showed a significantly (*p* < 0.05) higher value compared to both the HPM0 and HPM3 groups (53.5 ± 1.7, 53.5 ± 1.4, and 65.0 ± 4.1 for HPM0, HPM3, and HPM20, respectively). Finally, the HPM20 group showed a slight decrease in glycogen deposition compared to the other experimental groups (Figure 3d–f).

### 3.6. Spectroscopic Analyses

Figure 4 shows an example of FTIRI analysis on the liver samples obtained in the present study and highlights the topographical distributions of lipids, fatty acids, and glycogen as false-color images. To compare the biochemical composition of the liver samples belonging to different experimental diets, specific band area ratios were calculated and statistically analyzed (Figure 5). The HPM20 group showed significantly higher values of total lipids and fatty acids (LIP/TBM and FA/LIP; *p* < 0.05), and a significantly lower amount of glycogen (GLY/TBM; *p* < 0.05) compared to the other experimental groups, which did not show significant differences between them.

### 3.7. Real-Time PCR Results

Considering the expression of genes involved in the immune response, no significant differences were evident among the experimental groups (Figure 6).

## 4. Discussion

The physiological responses to different HPM dietary inclusion levels have been widely explored in several key fish species for the Mediterranean aquaculture, including rainbow trout [60,61] and gilthead seabream [62,63]. Considering the European seabass, most of the studies have focused on replacing FM with black soldier fly larvae meal rather than HPM. In particular, the available literature on European seabass highlighted that the dietary FM replacement with up to 50% black soldier fly larvae meal did not affect morphological traits, growth performance, somatic indexes, feed utilization, fillet quality, and hematological parameters of fish, leading to a potentiation of the oxidative status and an enhanced phagocytic activity [23,24,64]. Similarly, up to 45% of HPM can successfully replace FM in diets for European seabass juveniles without impairing growth, feed utilization, and digestibility [65]. Furthermore, Pleic et al. [26] recently demonstrated that, in diets deprived of FM, the replacement of plant-derived ingredients with defatted HPM, alone or in combination with poultry by-product meal, significantly improved the growth performance and led to positive effects on intestinal digestive/absorptive functions and microbial communities in European seabass subadults.

A desirable goal in modern aquaculture is the use of full-fat HPM, due to: (i) the high cost associated with the HPM defatting procedures; and (ii) the possibility to improve the quality of insects’ biomass by modulating the composition of their growth substrate with potential sources of long-chain PUFA using macroalgae or dried microbial biomass [32,34,66]. Accordingly, considering the promising results obtained in previous studies [35,39], the present research aimed, for the first time, to test two dietary substitution levels of FM with full-fat HPM obtained using a spirulina-enriched growth substrate on European seabass reared in aquaponic systems.

The European seabass represents one of the most important commercial species in the Mediterranean area and, as an euryhaline fish, has the potentiality to be reared in very-low-salinity aquaponic systems coupled with several commune plant species [67]. This approach can implement the fish farming production of coastal countries, especially considering the increasing limited water availability for aquaculture and agriculture. Thus, the suitability of this technique for European seabass rearing, demonstrated in previous studies and through the results of the present one, highlights the importance of a wider introduction of an environmentally sustainable method that is able to couple seafood production with minimal water consumption in modern aquaculture [67,68].

The dietary full-fat HPM inclusions tested in the present study did not have negative effects on fish survival and zootechnical performance, indicating that the experimental diets were well formulated and accepted by the fish. In addition, no behavioral alterations were observed among the experimental groups. These positive outcomes were expected, since no cognitive decline has been recorded in fish fed HPM-based diets (characterized by up to total FM replacement) [69], and most of the literature did not indicate a reduction in fish growth when the proportion of dietary HPM inclusion is lower than 30% [35,65,70,71]. Guaranteeing efficient growth performances using more sustainable aquafeed formulations, while ensuring fish welfare and quality represents a major challenge for the aquaculture sector. When HPM is included in fish diets, the application of a wide spectrum of laboratory techniques, with emphasis on gut/liver health status and marketable characteristics, are of primary importance due to the nutritional properties of this insect species [16,72].

In this context, the histological analyses performed in the present study did not show major changes in histological architecture or signs of inflammation in distal intestine samples from all the experimental groups. These results are supported by the molecular analyses of immune-response-related gene expression, which did not reveal significant differences among the experimental groups. These findings are in accord with previous studies that showed how even higher HPM dietary inclusion levels did not alter fish gut health status [28,73,74]. It is well known from the literature that HPM can have beneficial effects on this organ due to the presence of: (i) chitin, the main growth substrate of lactic acid bacteria able to make undigestible carbohydrates available and synthesize short-chain fatty acids with anti-inflammatory roles [75,76]; and (ii) anti-microbial peptides that enhance disease resistance against a wide range of pathogenic bacterial strains [24]. Furthermore, the gut health status and the absence of upregulation in immune response markers evidenced in the present study can also be related to a synergic cooperation of specific bioactive molecules present in the tested diets. In particular, lauric acid (C12), typically abundant in HPM fat fractions, and tocopherols and carotenoids (β-carotene and zeaxanthin), derived from spirulina, are well-known powerful molecules in preserving gut health status [37,77]. As expected, the only revealed change at gut level was an increase in Ab+ goblet cells abundance only in fish fed the HPM20 diet, possibly related to a higher lubrication of the intestine’s final tract related to the presence of undigested chitin [78].

Liver is one of the major storage sites of lipids, and particularly fatty acids, in many marine fish species, including European seabass [79]. The main drawback related to the use of full-fat HPM in aquafeeds is the unbalanced fatty acid profile that usually leads to an increase in SFA content and to a high n-6/n-3 ratio, which, in turn, has been demonstrated to promote hepatic lipid accumulation in fish [80]. Although the histological analysis of liver parenchyma in the present study did not show pathological alterations in any of the experimental groups, the percentage of fat fraction of HPM20 was significantly higher than in the other treatments, representing one more piece of evidence about the role of the HPM fat fraction in determining a high hepatic lipid accumulation. This result was also confirmed by FT-IR analyses on liver samples, which evidenced an increase in total lipids and in the fraction of fatty acids on total lipids in the HPM20 group. Furthermore, for this group, the hepatic energy storage content could mainly rely on fatty acids instead of glycogen, as evidenced by both the histological and spectroscopic analyses.

The fatty acid profile of full-fat HPM diets can also alter the fatty acid composition of the edible fish portion. Interestingly, a decrease in SFA content was observed from the HPM0 to HPM20 groups in the present study. Additionally, no significant changes in MUFA, n3, and n6 PUFA, as well as of important fatty acids, such as EPA and DHA, were evidenced in the fillets from fish fed an HPM20 diet compared to those of the HPM0 group. These results are consistent with a previous study on rainbow trout that used diets with similar inclusions of spirulina-enriched HPM [35]. Furthermore, a study by Moutinho et al. [23] did not show alterations in terms of the same fatty acid classes in fillets from European seabass fed on diets in which FM was replaced by up to 45% of black soldier fly larvae meal. As suggested by Moutinho et al. [23], the fillet fatty acid profile of lean fish, such as European seabass, is less influenced by the dietary lipid composition in comparison to fatter fish, such as salmonids, which are known to better reflect the unbalanced fatty acid profile of the HPM used for diet formulation [81,82,83]. In this context, it should also be considered that dietary characteristics can deeply influence the fillet oxidative stability, which, in turn, can be affected by the unsaturated/saturated fatty acid composition (with unsaturated fatty acids being more subjected to oxidation in comparison to saturated ones [84]) and by the dietary presence of antioxidant molecules [85]. As expected, the primary and secondary oxidative products were not significantly affected by the experimental diets in the present study for the above-mentioned reasons. Additionally, TBARS showed a slight decrease in the HPM3 and HPM20 groups compared to the HPM0 group, suggesting, as previously demonstrated, an involvement of α-tocopherol and carotenoids in preserving fillet quality features [39,86]. Finally, regarding the fillets’ physical properties, an alteration of the lightness (*L**), a particular feature that could influence consumer acceptance of the fish [87,88], has to be highlighted. This finding contrasts with previous studies that found *b** to be the most affected index by the addition of HPM in diets for rainbow trout [89], and with the results obtained by Moutinho et al. [23], which did not evidence significant differences in the visual properties of both skin and fillet of European seabass fed up to 45% of FM replacement with black soldier fly larvae meal.

## 5. Conclusions

The dietary inclusions of spirulina-enriched HPM tested in the present study ensured proper growth performance and did not cause behavioral alterations in European seabass juveniles. The bioactive molecules typical of HPM, such as chitin and lauric acid, as well as the ones derived from the spirulina dried biomass used to enrich the insects’ growth substrate (tocopherols and carotenoids) confirmed their key role in preserving the gut health status and the quality features of the fish edible fraction.

As a general conclusion, the results obtained in the present study highlighted the suitability of enriched full-fat HPM as a proper sustainable aquafeed ingredient for European seabass rearing and the use of aquaponic culturing systems to meet the requested criteria for a more sustainable aquaculture.

## Figures and Tables

**Figure 1 animals-13-01921-f001:**
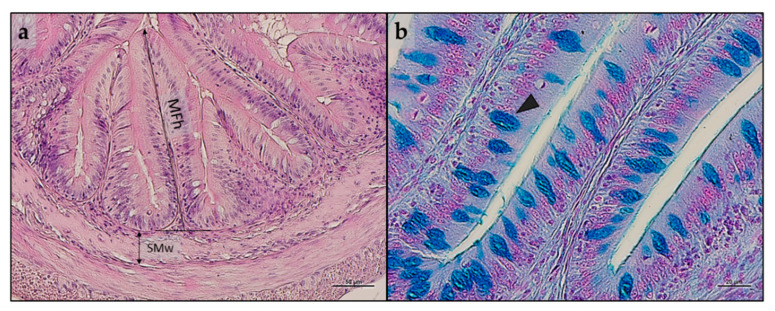
(**a**) Example of histomorphology of distal intestine from European seabass juveniles fed HPM0 diet with measurement criteria for the mucosal fold height (MFh) and submucosa width (SMw); (**b**) example of Ab + goblet cells (black arrowhead) detected in villus from fish fed HPM20 diet. Staining and scale bars: (**a**) Mayer’s hematoxylin and eosin Y, 50 µm; (**b**) Alcian blue, 20 µm.

**Figure 2 animals-13-01921-f002:**
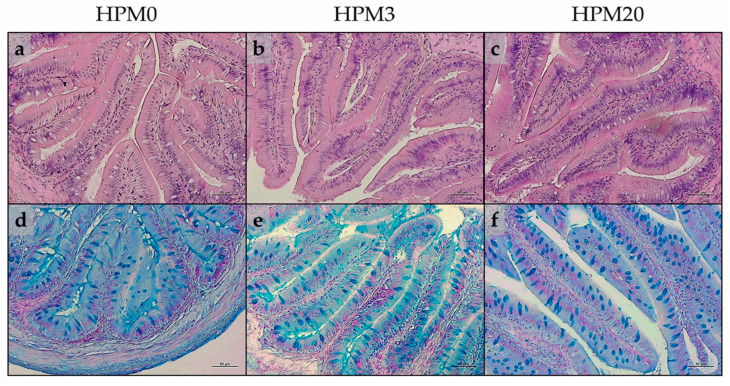
Example of histomorphology of distal intestine from European seabass juveniles fed the experimental diets. (**a**,**d**) HPM0; (**b**,**e**) HPM3; (**c**,**f**) HPM20. Stainings: (**a**–**c**) Mayer’s hematoxylin and eosin Y; (**d**–**f**) Alcian blue. Scale bars: 50 µm.

**Figure 3 animals-13-01921-f003:**
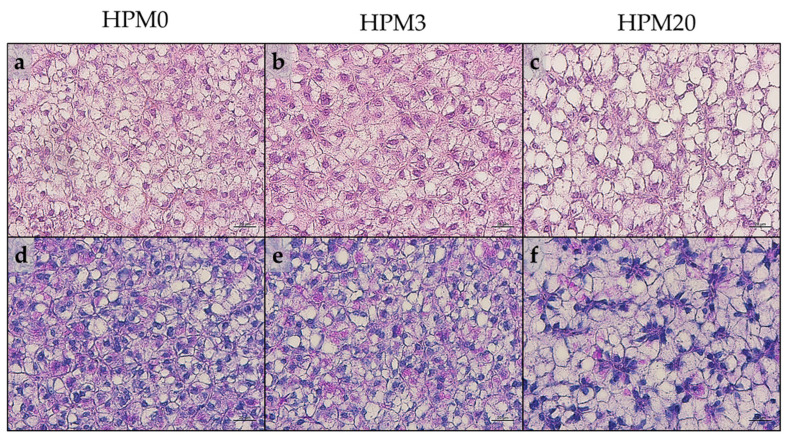
Example of histomorphology of liver from European seabass juveniles fed the experimental diets. (**a**,**d**) HPM0; (**b**,**e**) HPM3; (**c**,**f**) HPM20. Stainings: (**a**–**c**) Mayer’s hematoxylin and eosin Y; (**d**–**f**) PAS. Scale bars: 50 µm.

**Figure 4 animals-13-01921-f004:**
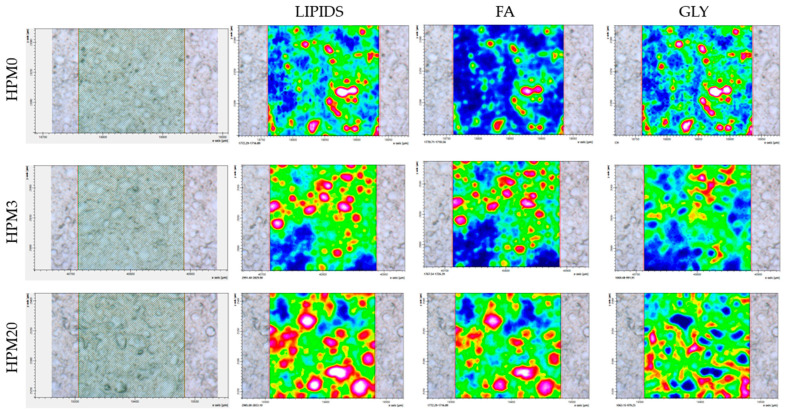
Microphotographs and false-color images of representative sections of HPM0, HPM3, and HPM20 liver samples showing the topographical distribution of lipids (LIP images, scale 0–2.5), of fatty acids (FA images, scale 0–1), and glycogen (GLY images, scale 0–5) on the mapped areas. IR maps are 164 × 164 μm in size and are composed by 4096 pixel/spectra with a spatial resolution 2.56 × 2.56 μm. A different color scale was adopted: white/light pink indicates high absorbance values of IR radiation, whilst black/dark blue indicates the low ones.

**Figure 5 animals-13-01921-f005:**
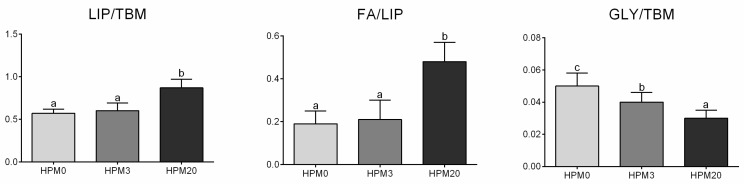
Biochemical composition of liver. Statistical analysis of the following band area ratios: LIP/TBM (relative amount of total lipids), FA/LIP (relative amount of fatty acids respect to total lipids), and GLY/TBM (relative amount of total glycogen). ^a–c^ Different letters indicate statistically significant differences.

**Figure 6 animals-13-01921-f006:**
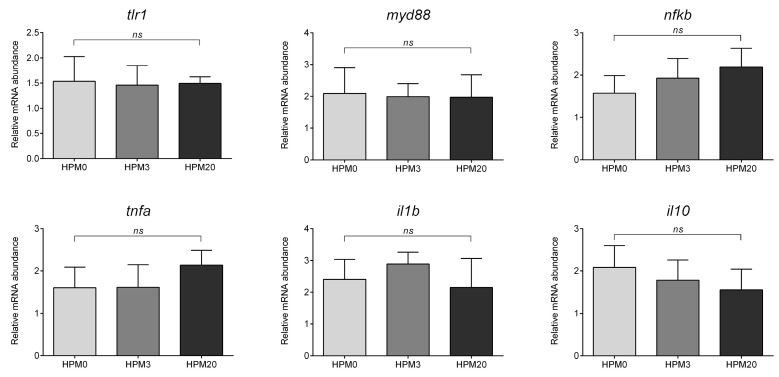
Relative mRNA abundance of genes involved in the immune response analyzed in distal intestine of European seabass juveniles fed the different experimental diets. Values are presented as mean ± SD (*n* = 15). ns, no significant differences (*p* > 0.05).

**Table 1 animals-13-01921-t001:** Ingredients (g/100 g), proximate composition (% as fed), and α-tocopherol and carotenoid contents of the experimental diets used in the present study.

	HPM0	HPM3	HPM20
**Ingredients (g/100g)**			
Fish meal ^1^	40.0	38.8	32.0
CPSP ^2^	5	5	5
Soy protein concentrate ^3^	11.1	11.1	13.0
HPM ^4^	-	1.2	8
Wheat gluten	11.1	11.1	13.0
Wheat bran	13.0	13.0	9.0
Starch, gelatinized	4	4	4
Fish oil	12.5	12.5	12.5
Soy lecithin	0.8	0.8	0.8
Mineral and Vitamin premix ^5^	1.5	1.5	1.5
DL-Methionine	-	-	0.1
L-Lysine			0.1
Binder ^6^	1	1	1
**Proximate composition (%)**			
Dry matter	93.1	93.1	93.3
Crude protein	47.4	47.1	47.2
Crude fat	17.6	17.6	17.8
Ash	9.1	9.0	8.8
Gross Energy (MJ/kg)	20.4	20.4	20.5
**α-tocopherol and carotenoid** **contents (mg/kg)**			
α-tocopherol	<LOD	2.04	11.12
β-carotene	<LOD	<LOD	5.98
zeaxanthin	<LOD	<LOD	2.25

^1^ Fish meal Super Prime (Pesquera Diamante Perù), CP 66%, CL 9%; ^2^ Fish Protein Concentrate Solubles (Sopropêche, Boulogne-sur-Mer, France); ^3^ Soy protein concentrate—Soycomil; ^4^ HPM prepupae, CP 36%, CL 13%; ^5^ Mineral and Vitamin premix (mg kg^−1^ diet): calcium carbonate, 1125; sodium chloride, 5640; potassium sulphate, 7500; magnesium sulphate, 3750; iron sulphate, 2640; manganese sulphate, 114; zinc sulphate, 207; copper sulphate, 60; cobalt chloride, 9.9; potassium iodide, 6; sodium selenite, 0.9; retinol, 18,000 (IU kg^−1^ diet); calciferol, 2000 (IU kg^−1^ diet); α-tocopherol, 35; menadione sodium bisulphite, 10; thiamin, 15; riboflavin, 25; Ca pantothenate, 50; nicotinic acid, 200; pyridoxine, 5; folic acid, 10; cyanocobalamin, 0.02; biotin, 1.5; ascorbic acid (stay C), 50; Inositol, 400; ^6^ Sodium alginate (Merck kGaA, Darmstadt, Germany). Values are reported as mean of duplicate analyses. LOD: limit of detection.

**Table 2 animals-13-01921-t002:** Score assignment criteria for mucosal fold fusion, basal inflammatory influx, supranuclear vacuolization of enterocytes, and goblet cell relative abundance in the distal intestine.

Parameter	Score	Description
Mucosal fold fusion	+	0–5 observations per section
	++	5–15 observation per section
	+++	>15 observation per section
Basal inflammatory influx	+	Scarce lymphocyte infiltration
	++	Moderated infiltration
	+++	Diffused infiltration
Supranuclear vacuoles	-	Absent
	+	Scattered
	++	Diffused
	+++	Highly abundant
Goblet cell relative abundance	+	Scattered cells
	++	Diffused and widely spread
	+++	Highly abundant and tightly packed cells

**Table 3 animals-13-01921-t003:** Primer sequences used in the present study, annealing temperature (A.T.), and NCBI IDs.

Gene	Forward Primer (5′–3′)	Reverse Primer (5′–3′)	A.T. (°C)	NCBI ID
*tlr1*	GGCCGACACACATACTTGGA	GCCTGGACGGTTTATGTGGT	59	KX_399287
*myd88*	CCGTCGTCTGTGGCTAACAT	GTCCCACGCCTTTTTCAACC	60	KM_225785
*nfkb*	GTTTGTCGTGTCGTTGGGAG	CGAGTGGACAAGTGAGTGGA	60	KM_225790
*il1b*	AACTCCAACAGCGCAGTACA	AGACTGGCTTTGTCCACCAC	58	AJ_311925
*il10*	GCAGTCCCATGTGCAACAAC	TGCTACTGAACCTACGTCGC	59	AM_268529
*tnfa*	GACTGGCGAACAACCAGATT	GTCCGCTTCTGTAGCTGTCC	59	DQ_070246
*b-actin* (hk)	GGTACCCATCTCCTGCTCCAA	GACGTCGCACTTCATGATGCT	60	AJ_537421
*18s* (hk)	AGGGTGTTGGCAGACGTTAC	CTTCTGCCTGTTGAGGAACC	60	XM_051390998

**Table 4 animals-13-01921-t004:** Survival rate and zootechnical performance of European seabass juveniles fed the experimental diets.

	HPM0	HPM3	HPM20	*p*-Value
IBW (g/fish)	19.3 ± 0.5	19.2 ± 0.5	19.4 ± 0.6	0.926
FBW (g/fish)	61.2 ± 4.3	60.3 ± 4.5	60.0 ± 5.1	0.659
RGR (%)	217.4 ± 22.3	214.3 ± 23.7	209.3 ± 26.5	0.551
SGR (% day^−1^)	1.47 ± 0.08	1.36 ± 0.09	1.34 ± 0.10	0.505
FCR	1.71 ± 0.09	1.75 ± 0.07	1.77 ± 0.11	0.728

Values are shown as mean ± SD (*n* = 3). Abbreviations: IBW, initial weight; FBW, final weight; WG, weight gain; RGR, relative growth rate; SGR, specific growth rate; FCR, feed conversion ratio.

**Table 5 animals-13-01921-t005:** Behavioral measurements during the open-field test conducted on European seabass juveniles fed the experimental diets.

	HPM0	HPM3	HPM20	*p*-Value
Distance covered (cm)	Q1 = 1538.0	Q1 = 647.3	Q1= 1155.0	0.292
	mdn = 2328.0	mdn = 1272.1	mdn = 3487.9	
	Q3 = 3038.0	Q3 = 2690.3	Q3 = 4962.5	
Immobility (s)	Q1 = 1072.4	Q1 = 1091.0	Q1 = 775.9	0.371
	mdn = 1098.4	mdn = 1117.0	mdn = 979.1	
	Q3 = 1141.2	Q3 = 1145.0	Q3 = 1129.8	
Time exploring the center (s)	Q1 = 0.24	Q1 = 0.00	Q1 = 2.40	0.096
	mdn = 1.40	mdn = 0.48	mdn = 8.80	
	Q3 = 4.00	Q3 = 2.85	Q3 = 34.40	
Turn angle (°)	Q1 = 71.18	Q1 = 67.09	Q1 = 67.05	0.762
	mdn = 74.38	mdn = 73.58	mdn = 70.15	
	Q3 = 76.45	Q3 = 78.54	Q3 = 93.56	
∆breathing frequency	0.85 ± 9.41	−4.92 ± 12.75	−6.72 ± 14.52	0.292

Values of ∆breathing frequency are shown as mean ± SD, while values of distance covered, immobility, time exploring the center of the arena, and turn angle are shown as median (mdn), 1st (Q1), and 3rd (Q3) quartile (*n* = 12).

**Table 6 animals-13-01921-t006:** Fatty acid profiles (% of total FAMEs) of the experimental diets.

	HPM0	HPM3	HPM20
C12:0	0.11 ± 0.01	0.85 ± 0.06	1.55 ± 0.12
C14:0	5.30 ± 0.13	5.38 ± 0.08	5.17 ± 0.24
C16:0	20.29 ± 0.24	20.70 ± 0.16	20.58 ± 0.69
C16:1-n7	6.61 ± 0.05	6.85 ± 0.01	6.77 ± 0.17
C18:0	4.27 ± 0.07	4.36 ± 0.12	4.43 ± 0.06
C18:1-n9(cis + trans)	16.09 ± 0.12	16.60 ± 0.30	16.73 ± 0.02
C18:1-n7	2.79 ± 0.01	2.88 ± 0.05	2.87 ± 0.01
C18:2-n6cis	9.09 ± 0.25	9.17 ± 0.03	9.46 ± 0.03
C18:3-n3	1.650 ± 0.010	1.580 ± 0.002	1.530 ± 0.026
C18:4-n3	1.47 ± 0.06	1.36 ± 0.02	1.26 ± 0.03
C20:1-n9	1.564 ± 0.030	1.619 ± 0.041	1.645 ± 0.003
C20:4-n6	1.06 ± 0.04	1.04 ± 0.02	0.98 ± 0.05
C20:5-n3	7.56 ± 0.19	7.05 ± 0.09	6.70 ± 0.29
C22:1-n11	0.98 ± 0.02	0.97 ± 0.01	1.04 ± 0.04
C22:5-n3	1.32 ± 0.01	1.21 ± 0.01	1.20 ± 0.06
C22:6-n3	11.99 ± 0.04	10.64 ± 0.31	10.18 ± 0.64
*Σ* *SFA*	32.09 ± 0.27	33.66 ± 0.13	34.38 ± 1.16
*Σ* *MUFA*	29.14 ± 0.17	30.00 ± 0.33	30.19 ± 0.08
*Σ n6* *PUFA*	11.19 ± 0.19	11.18 ± 0.06	11.40 ± 0.09
*Σ n3* *PUFA*	25.00 ± 0.32	22.77 ± 0.46	21.76 ± 1.11

SFA: saturated fatty acids; MUFA: monounsaturated fatty acids; PUFA: polyunsaturated fatty acids. For calculating the *Σ* classes, 47 fatty acids were used, but those with quantity below 1% of total FAMEs are not listed in the table.

**Table 7 animals-13-01921-t007:** Physical parameters of European seabass juveniles fed the experimental diets.

	HPM0	HPM3	HPM20	*p*-Value	RMSE
Standard length (cm)	16.61	16.39	16.03	ns	1.194
Total length (cm)	19.03	19.10	18.59	ns	1.274
Total weight (g)	63.86	65.94	61.57	ns	16.819
K	0.91	0.93	0.93	ns	0.101
Fillet yield	52.40 ^a^	52.53 ^a^	49.99 ^b^	0.016	2.683
Fillet weight (g)	16.71	17.33	15.65	ns	2.853
Intestine + Spleen (g)	0.68 ^b^	0.88 ^ab^	0.98 ^a^	0.036	0.324
Intestine + Spleen (%)	1.29 ^b^	1.68 ^ab^	1.93 ^a^	0.016	0.602
Residue (g)	28.60	40.09	27.80	ns	25.777
Residue (%)	54.70	75.69	55.27	ns	47.294
**Skin color**					
*L**	61.18 ^a^	58.14 ^b^	60.85 ^a^	0.005	2.725
*a**	−1.37	−1.47	−1.38	ns	0.271
*b**	−3.84	−3.32	−3.42	ns	1.190
**Fillet color**					
*L**	48.81 ^b^	48.43 ^b^	50.32 ^a^	0.001	0.858
*a**	−0.64	−0.76	0.11	ns	1.405
*b**	0.67	0.41	0.74	ns	0.394

K, condition factor; L*, lightness; a*, redness index; b*, yellowness index. RMSE: root mean square error. ^a^,^b^: different superscript letters indicate significant difference among groups (*p* < 0.05); ns: not significant differences.

**Table 8 animals-13-01921-t008:** Total lipids (g/100 g fresh tissue) and fatty acid profiles (mg of fatty acid/100 g fresh tissue) of fresh fillets from European seabass juveniles fed the experimental diets.

	HPM0	HPM3	HPM20	*p*-Value	RMSE
Total lipids	3.66	3.94	3.18	ns	0.801
**Fatty acids**					
C14:0	49.49	44.31	37.37	ns	14.224
C16:0	306.47 ^a^	287.73 ^a^	213.25 ^b^	0.031	68.638
C16:1n-7	76.73	83.23	67.09	ns	25.296
C18:0	72.18 ^a^	65.99 ^a^	50.96 ^b^	0.016	13.705
C18:1n-9	309.06	382.52	280.30	ns	89.191
C18:1n-7	43.02	48.62	38.39	ns	12.508
C18:2n-6	155.16 ^b^	213.65 ^a^	155.81 ^b^	0.034	47.609
C18:3n-3	23.63	30.27	21.82	ns	7.345
C20:1n-9	34.60	34.98	29.61	ns	8.406
C20:5n-3	92.83	90.79	77.43	ns	25.370
C22:1n-11	21.92	20.15	18.26	ns	5.226
C22:6n-3	217.67	218.51	178.84	ns	45.818
*Σ SFA*	449.60 ^a^	416.20 ^a^	319.03 ^b^	0.046	101.535
*Σ MUFA*	504.57	589.92	451.19	ns	142.574
*Σ n6 PUFA*	189.62 ^b^	251.00 ^a^	187.29 ^b^	0.054	55.693
*Σ n3 PUFA*	374.18	379.25	311.39	ns	89.725

SFA: saturated fatty acids; MUFA: monounsaturated fatty acids; PUFA: polyunsaturated fatty acids. For calculating the Σ, 47 fatty acids were utilized, but those with a quantity below 30 mg/100 g of fresh tissue, corresponding to 1% of total FAME, are not listed in the table. RMSE: root mean square error; ns: no significant differences. ^a^,^b^: different superscript letters indicate significant difference among groups (*p* < 0.05).

**Table 9 animals-13-01921-t009:** Conjugated dienes (CD, mmol kg^−1^ fresh tissue) and thiobarbituric-acid-reactive substances (TBARS, mg MDA-eq. kg^−1^ fresh tissue) of European seabass juveniles fed the experimental diets.

	HPM0	HPM3	HPM20	*p*-Value	RMSE
**Fillets**					
CD	0.13	0.14	0.12	ns	0.029
TBARS	0.04	0.03	0.03	ns	0.020

RMSE, root mean square error; ns, no significant differences.

**Table 10 animals-13-01921-t010:** Histological indexes measured in distal intestine of European seabass juveniles fed experimental diets.

	HPM0	HPM3	HPM20
Mucosal fold height	458.7 ± 22.3	449.5 ± 12.9	463.0 ± 13.5
Submucosa width	26.2 ± 2.7	25.9 ± 1.3	27.1 ± 2.7
Mucosal fold fusion events	+	+	+
Basal inflammatory influx	+	+	+
Supranuclear vacuoles	-	-	-
Ab+ goblet cells relative abundance	++	++	+++

Values for mucosal fold height and submucosa width are shown as mean ± SD (*n* = 30).

## Data Availability

The data presented in this study are available on request from the corresponding author.

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
