# Peer review of "Enhancing Rearing of European Seabass (Dicentrarchus labrax) in Aquaponic Systems: Investigating the Effects of Enriched Black Soldier Fly (Hermetia illucens) Prepupae Meal on Fish Welfare and Quality Traits"

_animals, 2023, doi:10.3390/ani13121921_

Round 1

Reviewer 1 Report

The study used a multidisciplinary methodology, encompassing zootechnical parameters, marketable quality traits, physiological and behavioral responses was to assess fish fed on diets including enriched black soldier fly quality and welfare. I think this article is well designed and well written, and I agree to accept it, but just have a question:

Why choose those two supplementation ratios of 3 and 20% in this study.

I think author should supplement the reason in line 146.

I think this article is well written.

Author Response

Rev 1

The study used a multidisciplinary methodology, encompassing zootechnical parameters, marketable quality traits, physiological and behavioral responses was to assess fish fed on diets including enriched black soldier fly quality and welfare. I think this article is well designed and well written, and I agree to accept it, but just have a question:

Why choose those two supplementation ratios of 3 and 20% in this study.

I think author should supplement the reason in line 146.

Comments on the Quality of English Language

I think this article is well written.

Thank you for the suggestion. We modified the sentence at lines 151-155 of the revised MS, to include the reason under the choice of the FM replacement levels used in the present study, as follows: “Particularly, a control diet (HPM0) based on conventional marine (FM and fish oil) and vegetable ingredients and two diets obtained from the control one by replacing FM at 3 % (as a feed supplement inclusion) or 20 % (this dietary FM replacement level in aquafeeds at global level has a significant ecological impact) with spirulina-enriched HPM (named HPM3 and HPM20, respectively) were produced”

Reviewer 2 Report

Why did the authors not provide FCR values for the tested feeds?

Author Response

Rev 2

Comments and Suggestions for Authors

Why did the authors not provide FCR values for the tested feeds?

Thank you for the suggestion. We agree that FCR represents an important additional information regarding the zootechnical performance obtained using the experimental diets. For that reason, we included this result in Table 3 and we specified the formula in the paragraph 2.4.

Reviewer 3 Report

The study presented valuable insights into the inclusion of full-fat Hermetia illucens prepupae meal (HPM) in European seabass diets within aquaponic systems. The researchers found that HPM did not negatively affect fish survival, growth performance, immune response markers, or gut health status. The study's findings contribute to the development of sustainable aquafeed formulations and promote the circular economy concept in aquaculture.

The title could be slightly rearranged to improve readability, for instance, something like, "Enhancing Rearing of European Seabass (Dicentrarchus labrax) in Aquaponic Systems: Investigating the Effects of Enriched Black Soldier Fly (Hermetia illucens) Prepupae Meal on Fish Welfare and Quality Traits".

A couple of general comments:

There are quite a bit of long sentences that could be broken down into smaller sentences for better clarity, a prime example is in the Abstract- lines 41 to 45. Also, be consistent with certain terms, for instance, either use "aquaponics" or "aquaponic system"

A couple of corrections:

Line 28- "exploit the application of" could be replaced with "utilize" or "apply," for a clearer, more positive meaning, as exploit tends to have a negative connotation. 

Line 53- "The human population is expected to reach 8.5 billion by 2030 and 9.7 billion by 2050" would sound better.

Line 138- "continuous"

Line 219- "feed"

Line 255- "at the end of the test"

Line 549- I suggest "most of the studies have focused on replacing FM with black soldier fly larvae meal rather than HPM."

Line 571- I suggest "The inclusion of full-fat HPM in the diets did not have any negative effects on fish survival and growth performance, indicating that the experimental diets were well-formulated and accepted by the fish."

Line 586- I suggest "These findings are supported by molecular analyses of immune response-related gene expression, which did not reveal significant differences among the experimental groups."

Line 606- I suggest "Although the histological analysis of liver parenchyma in the present study did not show any pathological alterations in any of the experimental groups..."

Line 616- I suggest "Interestingly, a decrease in saturated fatty acid (SFA) content was observed from the HPM0 to HPM20 groups in the present study."

Line 619- I suggest "These results are consistent with a previous study on rainbow trout that used diets with similar inclusions of spirulina-enriched HPM."

Line 631- I suggst "As anticipated, the primary and secondary oxidative products were not significantly affected by the experimental diets in the present study due to the composition of the diets and the presence of antioxidant molecules." or please modify to state the reasons.

Line 638- I suggest "This finding contrasts with previous studies that found b* to be the most affected index by the addition of HPM in diets for rainbow trout."

In the abstract, please specify the quantities in the key findings of the study:

Emphasize the survival and growth rates, absence of behavioural alterations, and the positive impact on gut health.

Highlight that the experimental diets did not negatively affect the fillet fatty acid profile or physical features, and prevented lipid oxidation.

In the Introduction, please:

Emphasize the need for sustainable aquafeed formulations and culturing systems to meet sustainability criteria.

Elaborate a little on the use of aquaponics for water conservation and waste reuse.

In the Materials and Methods section, please:

Briefly outline the protocol for alpha-tocopherol and carotenoid analyses

In Table 1, there is "<LOD", is that for "Limit of Detection"? If yes, please state it.

In the Discussion section, please:

Attempt to elaborate on the potential implications of using full-fat HPM in aquafeeds

Consider discussing the economic benefits and the potential to enhance the quality of insect biomass through growth substrate modulation

Provide further insights into the advantages and challenges associated with the use of full-fat HPM in sustainable aquaculture practices, highlighting the importance of considering factors such as fish welfare, gut health, and fillet properties

Mention the significance or role of the aquaponic element of the research, if any. 

Please re-read your draft and address some of the suggestions made above.

Author Response

Rev 3

The study presented valuable insights into the inclusion of full-fat Hermetia illucens prepupae meal (HPM) in European seabass diets within aquaponic systems. The researchers found that HPM did not negatively affect fish survival, growth performance, immune response markers, or gut health status. The study's findings contribute to the development of sustainable aquafeed formulations and promote the circular economy concept in aquaculture.

The title could be slightly rearranged to improve readability, for instance, something like, "Enhancing Rearing of European Seabass (Dicentrarchus labrax) in Aquaponic Systems: Investigating the Effects of Enriched Black Soldier Fly (Hermetia illucens) Prepupae Meal on Fish Welfare and Quality Traits".

We agree with the Reviewer and we decided to modify the title according to the suggestion.

A couple of general comments:

There are quite a bit of long sentences that could be broken down into smaller sentences for better clarity, a prime example is in the Abstract- lines 41 to 45. Also, be consistent with certain terms, for instance, either use "aquaponics" or "aquaponic system"

Some sentences have been divided as suggested by the Reviewer and we checked for terms that were not consistent along the text.

A couple of corrections:

Line 28- "exploit the application of" could be replaced with "utilize" or "apply," for a clearer, more positive meaning, as exploit tends to have a negative connotation. 

Line 53- "The human population is expected to reach 8.5 billion by 2030 and 9.7 billion by 2050" would sound better.

Line 138- "continuous"

Line 219- "feed"

Line 255- "at the end of the test"

Line 549- I suggest "most of the studies have focused on replacing FM with black soldier fly larvae meal rather than HPM."

Line 571- I suggest "The inclusion of full-fat HPM in the diets did not have any negative effects on fish survival and growth performance, indicating that the experimental diets were well-formulated and accepted by the fish."

Line 586- I suggest "These findings are supported by molecular analyses of immune response-related gene expression, which did not reveal significant differences among the experimental groups."

Line 606- I suggest "Although the histological analysis of liver parenchyma in the present study did not show any pathological alterations in any of the experimental groups..."

Line 616- I suggest "Interestingly, a decrease in saturated fatty acid (SFA) content was observed from the HPM0 to HPM20 groups in the present study."

Line 619- I suggest "These results are consistent with a previous study on rainbow trout that used diets with similar inclusions of spirulina-enriched HPM."

Line 631- I suggst "As anticipated, the primary and secondary oxidative products were not significantly affected by the experimental diets in the present study due to the composition of the diets and the presence of antioxidant molecules." or please modify to state the reasons.

Line 638- I suggest "This finding contrasts with previous studies that found b* to be the most affected index by the addition of HPM in diets for rainbow trout."

All the sentences have been modified.

In the abstract, please specify the quantities in the key findings of the study:

Emphasize the survival and growth rates, absence of behavioural alterations, and the positive impact on gut health.

Highlight that the experimental diets did not negatively affect the fillet fatty acid profile or physical features, and prevented lipid oxidation.

Unfortunately, we cannot emphasized more the content of the abstract since it reaches the maximum number of words provided by the authors guidelines. We think that all the key results are well depicted in the abstract. However, to try to agree with the reviewer suggestion,  we modified some sentences in order to highlight the positive results obtained and to add some values.

In the Introduction, please:

Emphasize the need for sustainable aquafeed formulations and culturing systems to meet sustainability criteria.

Elaborate a little on the use of aquaponics for water conservation and waste reuse.

Both these aspects have been elaborated in the last part of introduction (lines 109-124) of the revised MS.

In the Materials and Methods section, please:

Briefly outline the protocol for alpha-tocopherol and carotenoid analyses

In Table 1, there is "<LOD", is that for "Limit of Detection"? If yes, please state it.

A brief protocol for alpha-tocopherol and carotenoids determination has been added in lines 184-197 of the revised MS. In addition, “LOD: limit of detection” has been added in the caption of the Table 1.

In the Discussion section, please:

Attempt to elaborate on the potential implications of using full-fat HPM in aquafeeds

This point has been emphasized in lines 636-639 of the revised MS.

Consider discussing the economic benefits and the potential to enhance the quality of insect biomass through growth substrate modulation

This point has been emphasized in lines 583-587 of the revised MS

Provide further insights into the advantages and challenges associated with the use of full-fat HPM in sustainable aquaculture practices, highlighting the importance of considering factors such as fish welfare, gut health, and fillet properties

We appreciate the comment, but we think that this point doesn’t need further details since the results obtained are linearly discussed and supported by the available literature on this topic. We think that all these aspects are already present in the MS.

Mention the significance or role of the aquaponic element of the research, if any. 

This point has been emphasized in lines 593-600 of the revised MS

The authors would like to thank the reviewer for the comments. We tried to satisfy all the suggestions without applying deep modifications to the text, also in light to the absence of revising request from the other  2 reviewers. We hope that the modifications applied will satisfy the reviewer requests.